**PLOS** NEGLECTED TROPICAL DISEASES

# Old tools, new applications: Use of environmental bacteriophages for typhoid surveillance and evaluating vaccine impact

Yogesh Hooda[1], Shuborno Islam[1], Rathin Kabiraj[1], Hafizur Rahman[1], Himadree Sarkar[1], Kesia E. da Silva[2], Rajan Saha Raju[1], Stephen P. Luby[2], Jason R. Andrews[2], Samir K. Saha[1,3], Senjuti Saha [1] *

**1** Child Health Research Foundation, Dhaka, Bangladesh, **2** Division of Infectious Diseases and Geographic Medicine, Stanford University School of Medicine, Stanford, California, United States of America, **3** Department of Microbiology, Bangladesh Shishu Hospital and Institute, Dhaka, Bangladesh

* senjutisaha@chrfbd.org

**Data Availability Statement:** All data is present in the manuscript and its supporting files.

**Funding:** The present study was the funded by a grant from the Bill and Melinda Gates Foundation

## Abstract

Typhoid-conjugate vaccines (TCVs) provide an opportunity to reduce the burden of typhoid fever, caused by *Salmonella* Typhi, in endemic areas. As policymakers design vaccination strategies, accurate and high-resolution data on disease burden is crucial. However, traditional blood culture-based surveillance is resource-extensive, prohibiting its large-scale and sustainable implementation. *Salmonella* Typhi is a water-borne pathogen, and here, we tested the potential of Typhi-specific bacteriophage surveillance in surface water bodies as a low-cost tool to identify where *Salmonella* Typhi circulates in the environment. In 2021, water samples were collected and tested for the presence of *Salmonella* Typhi bacteriophages at two sites in Bangladesh: urban capital city, Dhaka, and a rural district, Mirzapur. *Salmonella* Typhi-specific bacteriophages were detected in 66 of 211 (31%) environmental samples in Dhaka, in comparison to 3 of 92 (3%) environmental samples from Mirzapur. In the same year, 4,620 blood cultures at the two largest pediatric hospitals of Dhaka yielded 215 (5%) culture-confirmed typhoid cases, and 3,788 blood cultures in the largest hospital of Mirzapur yielded 2 (0.05%) cases. 75% (52/69) of positive phage samples were collected from sewage. All isolated phages were tested against a panel of isolates from different *Salmonella* Typhi genotypes circulating in Bangladesh and were found to exhibit a diverse killing spectrum, indicating that diverse bacteriophages were isolated. These results suggest an association between the presence of Typhi-specific phages in the environment and the burden of typhoid fever, and the potential of utilizing environmental phage surveillance as a low-cost tool to assist policy decisions on typhoid control.

## Author summary

The WHO prequalified two typhoid conjugate vaccines for use to reduce the burden of typhoid fever. As policymakers design vaccination strategies, accurate, high-resolution estimates of typhoid burden are crucial for efficient use of the vaccines. Typhoid burden

(INV003717) to SS. The funders had no role in study design, data collection and analysis, decision to publish, or preparation of the manuscript.

**Competing interests:** The authors have declared that no competing interests exist.

can vary widely; for example, in Bangladesh, burden is high in the urban capital city Dhaka, but is 100-fold lower in the rural site, Mirzapur. Optimal local data rely on traditional blood-culture-based surveillance, which is expensive and often unavailable. With the knowledge that *Salmonella* Typhi, cause of typhoid fever, is a water-borne pathogen, we tested an environmental surveillance tool that detects bacteriophages (viruses) against *Salmonella* Typhi in environmental water bodies using simple assays. Testing of 303 water samples from Dhaka and Mirzapur showed a 10-fold lower abundance of bacteriophages in Mirzapur, depicting a correlation with typhoid burden in the community. This low-cost surveillance can be employed in different regions to generate rapid data on typhoid burden for evidence-based introduction of vaccines and tracking their impact upon rollout.

## Introduction

Typhoid fever is a systemic infection caused by the water-borne pathogen *Salmonella enterica* serovar Typhi. This pathogen is common in many low- and middle-income countries (LMICs) causing an estimated 135,000 deaths and 14 million infections globally [1]. The World Health Organization (WHO) recommends the use of typhoid conjugate vaccines in settings with high burden of typhoid fever [2,3]. Countries have begun implementing these suggestions, and several countries have performed large-scale clinical trials to demonstrate that these vaccines exhibit 80–85% efficacy in preventing infections [4–6]. Typhoid-conjugate vaccines are currently being utilized to tackle an extensively drug-resistant (XDR) *Salmonella* Typhi outbreak in Hyderabad, Pakistan, where vaccine effectiveness has been demonstrated to be high [7].

Decisions are currently being made to roll out typhoid-conjugate vaccines in other endemic countries, however, accurate and high-resolution spatial data regarding the burden of typhoid is required for optimal use of the available vaccines. The current estimates of burden of typhoid fever in countries have primarily come from modelling studies based on limited surveillance data and do not provide the geographical and temporal resolution within local communities and countries to design effective preventive and treatment measures. The paucity of the data is in large part because traditional blood culture surveillance is resource extensive, requires clinical laboratory infrastructure and trained health and research professionals. Consequently, very few LMICs routinely conduct these studies, and a few that do, tend to focus on high-risk urban settings, and are not able to sustain these studies as a regular part of health infrastructure. This has led to search for low-cost and sustainable methods to supplement traditional clinical surveillance systems [8,9]. To this end, environmental surveillance strategies that can identify *Salmonella* Typhi in different water supply have been proposed. In recent years, environmental wastewater surveillance has been used for early detection and monitoring the spread of SARS-CoV-2 [10,11] and poliovirus [12] among others.

Previous studies have shown that high detectable levels of *Salmonella* Typhi in the water supply overlaps with areas of disease burden, suggesting sampling water could be utilized as a preliminary surveillance proxy [8,13]. While it has not been possible to reproducibly culture *Salmonella* Typhi directly from environmental water, quantitative polymerase chain reaction (qPCR) has been proposed to detect *Salmonella* Typhi DNA [9,14]. However, qPCR-based methods cannot be replicated in most settings with typhoid burden, due to the lack of infrastructure (machines, molecular techniques etc.) and the associated high costs. To address this

gap, we investigated if the presence of bacteriophages in the environmental water samples could be used as a proxy to estimate the prevalence of *Salmonella* Typhi in water bodies.

Bacteriophages (or phages) are viruses that infect bacteria and bacterial abundance, phenotypic characteristics, and long-term evolutionary trajectory. Bacteriophages are very specific to their host bacterial species and often able to discriminate between different sub-populations based on minor genetic differences in host receptor and surface epitopes [15]. Bacteriophages against *Salmonella* Typhi (Typhi phages) were first reported in the 1940's [16]. Typhi phages were extensively used for bacterial typing in the 1940s-80s before the advent of molecular diagnostic methods such as PCR and genomic sequencing [17]. However, little has been done in the last 50 years and there is a lack of contemporary literature regarding Typhi phages from typhoid-endemic countries. This motivated us to initiate a pilot study to determine the feasibility of detecting *Salmonella* Typhi-specific phages in water bodies and typhoid burden at two geographic regions: Dhaka, a city of 9 million people with a high typhoid burden and Mirzapur, a rural district with 340,000 people that has low typhoid burden. Overall, our work serves as a pilot study aimed at investigating the potential correlation between the prevalence of Typhi bacteriophages and the local typhoid burden.

## Methods

### Ethics statement

The protocols of this study were approved by the ethics review committee of the Bangladesh Institute of Child Health, BSHI. For the hospitalized cases, informed written consent were taken from parents/legal guardians of all participants.

### Data from clinical surveillance and ethical considerations

The clinical laboratories of Bangladesh Shishu Hospital and Institute [BSHI], Shishu Shasthya Foundation Hospital [SSFH], and Kumudini Women Medical College and Hospital [KWMCH] are part of the laboratory network of Child Health Research Foundation CHRF, where all data are stored electronically. These laboratories are part of the WHO-supported Invasive Bacterial Vaccine Preventable Surveillance conducted at the CHRF and has been described earlier [18]. Data on blood culture surveillance was obtained from these electronic records.

### Environmental sample collection and bacteriophage isolation

Water samples were collected from sewage drains, rivers, ponds, lakes, and stagnant water bodies selected based on accessibility. Stagnant water bodies are defined as temporary water bodies that form after flooding or rainfall that cannot be used for livestock or human use, often due to poor water quality. A sterile cup, attached to a rope, was used to collect >10 ml of water sample from each source. Maintaining sterile techniques, the sample was transferred into a sterile bottle for transportation. Ten ml of the sample was centrifuged at 500x g for 5 minutes in a 15 ml conical tube to pellet large soil debris. The supernatant was passed through a 0.22 μm PES syringe filter into a new tube. The filtered sample was stored at 4˚C up to 72 hours before use for further experiments.

A total of 500 μl of the filtered water sample was mixed with 450 μl of LB broth and 50 μl of overnight liquid *Salmonella* Typhi strain BRD948 culture in a 2 ml microcentrifuge tube. *Salmonella* Typhi strain BRD948 is an attenuated strain derived from the laboratory strain Ty2, allowing for processing of the samples in a Biosafety Level-2 facility. The strain does not contain any antibiotic resistance plasmids and is thus susceptible to most antibiotics. This mixture

was incubated at 37˚C for 2 hours followed by the addition of 2–3 drops of chloroform and vortexing to lyse and kill all bacteria in the mixture without affecting bacteriophages. The mixture was then centrifuged at 10,000x g for 10 minutes and 750 μl of the supernatant potentially containing enriched *Salmonella* Typhi bacteriophages was transferred to a new tube.

## Bacteriophage detection & propagation

For bacteriophage detection, the enriched sample was tested using the double-layer agar method described earlier [19]. In brief, 100 μl of the sample was incubated with 200 μl of overnight culture of *Salmonella* Typhi strain BRD948 for 20 minutes. The entire 300 μl was mixed with 4 ml of molten soft agar (0.7% agar) and poured over solid hard agar plates (1.5% agar). These plates were incubated at 37˚C for 14–16 hours and the appearance of plaques the next day indicated the presence of bacteriophages in the sample. For each plaque morphology, a plaque was picked using a 100 μl pipette tip and resuspended in 100 μl of tryptic soy broth. Two drops of chloroform were added to the resuspension followed by 10 minutes of incubation, and a final centrifugation at 10,000x g for 10 minutes. 70 μl of the supernatant was transferred to a new tube which contained a clone of phages of a single morphology. To calculate phage titers for each isolated phage (in plaque forming units or pfu), we spotted 2 μl of the supernatant at $10^0$, $10^2$, $10^4$ and $10^6$ dilutions on a lawn of *Salmonella* Typhi strain BRD948.

## Activity spectra of Typhi phages

To confirm that these phages are Typhi specific, 2 μl of at least $3.5 * 10^3$ pfu/ml of isolated phages were spotted on strains of *Salmonella enterica* serovars Paratyphi A (E321: 1100582310), Paratyphi B (366817) and Typhimurium LT2, and Gram-negative bacteria *Escherichia coli* (ATCC: 25922), *Klebsiella pneumoniae* (ATCC: 700603 & ATCC: 9349) and *Pseudomonas aeruginosa* (ATCC: 27853) using the double layer methodology.

To test diversity of the phages, 2 μl of at least $3.5 * 10^3$ pfu/ml of isolated phages were also spotted on 16 *Salmonella* Typhi isolates collected from blood culture belonging to different genotypes previously described to be present in Bangladesh [20]. The plates were incubated overnight at 37˚C for 14–16 hours and the plates were visualized for zones of clearing. Killing activities against the different genotypes were recorded and hierarchical clustering and heat map was generated using the R package *gplots*.

## RFLP assay

To determine diversity of the phages, 20 of the 86 phages representing different killing spectra were selected for restriction fragment length polymorphism (RFLP). First, 350 pfu/ml of each phage sample was incubated with 200 μl of overnight culture of *Salmonella* Typhi strain BRD948 for 20 minutes, and then mixed with 4 ml of molten soft agar and poured over solid hard agar plates to obtain confluent lysis. The plates were incubated at 37˚C overnight [21]. The next day, 4 ml of LB was added to each plate and incubated at 4˚C for 4 hours to elute phages from the top layer. The liquid was transferred to a 15 ml tube, followed by the addition of 4 drops of chloroform and vortexing. The mixture was centrifuged at 10,000x g for 10 minutes and the supernatant was transferred to a new tube.

To remove any residual bacterial DNA and RNA present in the lysate, 450 μl of the supernatant was incubated with 50 μl DNase I 10x buffer, 1 μl DNase I (1 U/μl), and 1 μl RNase A (10 mg/ml) for 1.5 h at 37˚C without shaking. 20 μl of 0.5 M EDTA (final concentration 20 mM) was added to inactivate DNase I and RNase A after incubation. To digest phage protein capsids, 1.25 μl of Proteinase K (20 mg/ml) was added and incubated for 1.5 h at 56˚C without shaking. DNA extraction was conducted using the DNAeasy Blood and Tissue Kit (Qiagen,

69504) using Manufacturer's instructions. 5 μl of extracted DNA was run on a 1% agarose gel to visualize the purified phage DNA. The concentration and quality of the DNA was assessed using the Nanodrop.

200 ng DNA were digested with NheI (10 U/μl) and XbaI (15 U/μl) restriction enzymes in a single 10 μl reaction for 1 hr at 37°C, followed by heat inactivation at 65°C for 20 mins. The different DNA fragments were visualized by running 5 μl of digested DNA on a 0.8% agarose gel. The gel image was analyzed using Fiji [22] to identify different gel bands and the molecular weights of each band was estimated using a 1kb Plus DNA ladder (ThermoFisher Scientific, 10787026) as reference.

## Results

### Clinical typhoid surveillance

Since 2012, we have been conducting surveillance to monitor enteric fever, pneumonia, meningitis, and sepsis (as part of the Invasive Bacterial Vaccine Preventable Disease Surveillance of the WHO) in two hospitals in urban Dhaka (BSHI and SSFH), and in one hospital in Mirzapur (KWMCH), a rural district approximately 60 km north of Dhaka [18,23]. In 2021, we performed 4,620 blood cultures in BSHI and SSFH, of which 215 (4.7%) were culture-confirmed for *Salmonella* Typhi. In contrast, at KWMCH, during the same period 3,788 blood cultures were performed and only 2 (0.05%) were culture-confirmed for *Salmonella* Typhi (Table 1). The burden of culture-confirmed cases of typhoid fever in Mirzapur is 100-fold lower.

### Environmental phage surveillance

Between August 2021 and December 2021, we collected 211 environmental water samples in the Dhaka region. These samples constituted of sewage water (n = 168), lake (n = 18), pond (n = 15), river (n = 5) and stagnant water (n = 5) from different sites across Dhaka (Fig 1A). Sixty-six of the 211 samples (31%) exhibited plaque formation, and in all cases, the plaques could be propagated confirming the presence of active bacteriophages. We observed at least two morphologies in 16 samples suggesting different phages in the same sample. Phages of different morphologies could be further purified during propagation bringing the total number

**Table 1. Blood culture positivity of *Salmonella* Typhi in clinical surveillance, and bacteriophage positivity in environmental water samples in Dhaka and Mirzapur, Bangladesh.**

| Study site in Bangladesh | Type of surveillance | Type of sample | Total no. of samples tested | No. of *Salmonella* Typhi/phage positive samples | % Positive samples |
|---|---|---|---|---|---|
| Dhaka | Clinical surveillance | Blood | 4620 | 215 | 5% |
| | Environmental surveillance | Sewage | 168 | 51 | 30% |
| | | Lake | 18 | 10 | 56% |
| | | Pond | 15 | 2 | 13% |
| | | River | 5 | 2 | 40% |
| | | Stagnant water | 5 | 1 | 20% |
| | | Total | 211 | 66 | 31% |
| Mirzapur | Clinical surveillance | Blood | 3788 | 2 | 0.05% |
| | Environmental surveillance | Sewage | 3 | 1 | 33% |
| | | Pond | 48 | 0 | 0% |
| | | River | 4 | 0 | 0% |
| | | Stagnant water | 37 | 2 | 5% |
| | | Total | 92 | 3 | 3% |

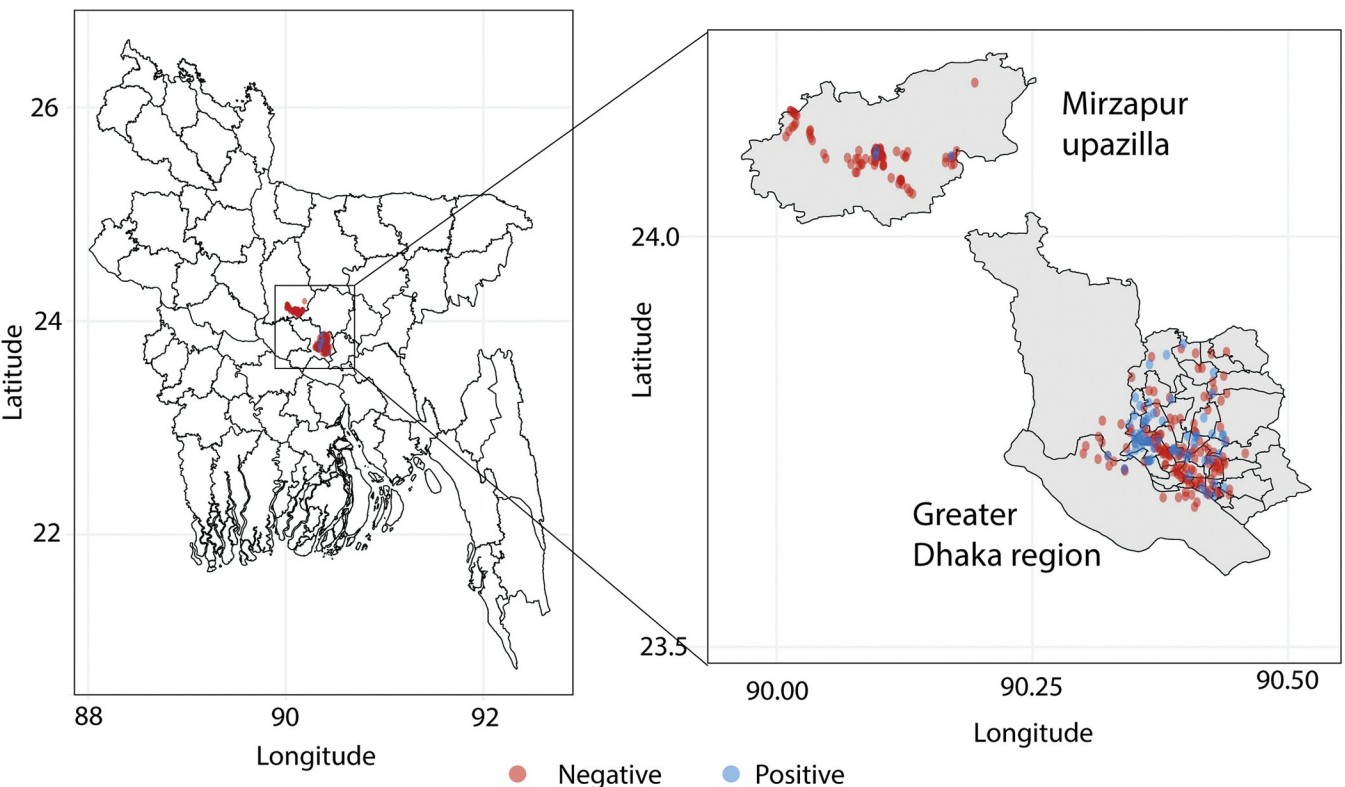

**Fig 1. Location of water sample collection and detection of *Salmonella* Typhi phages in water samples.** Water sampling was conducted in two districts in Bangladesh: urban Dhaka and rural Mirzapur, Bangladesh. The water bodies sampled where Typhi-bacteriophages were detected are shown in blue while the water bodies where no Typhi-bacteriophages were found are shown in red. The map was obtained from the database of Global Administrative Areas (GADM), a free and open-source database (license: https://gadm.org/license.html, CC BY) through the R packages *maps* and *maptools*.

of isolated lytic phages to 82 from 211 samples in Dhaka. The distribution of positive and negative samples showed that Typhi phages were present in all types of water bodies tested (Table 1).

In contrast, a total of 92 environmental samples were collected during the same time from the Mirzapur region (Fig 1B). These samples constituted of sewage water (n = 3), pond (n = 48), river (n = 4), and stagnant water (n = 37). A total of 3 samples showed positive phage lytic activity in the 92 samples tested (3%), the details of which are provided in Table 1. Two different plaque morphologies were noted in one sample, bringing a total of 4 isolated phages from these 92 samples (Table 1). No phages were detected in ponds or rivers; one positive sample was from sewage water (n = 1, 33%) and two from stagnant water. Overall, phage prevalence in water bodies in Dhaka (31%) was 10-fold higher than that in water bodies in Mirzapur (3%), correlating with the culture-confirmed typhoid cases in the largest hospitals of the regions.

## Host range and diversity of Typhi phages

To test the specificity of phages isolated, we tested all 86 isolated phages against closely related *Salmonella enterica* serovars Typhimurium, Paratyphi A and Paratyphi B, and closely related *Enterobacteriaceae* species *Escherichia coli* and *Klebsiella pneumoniae*. Another gamma-proteobacterium *Pseudomonas aeruginosa* was also included. No cross activity was observed, depicting that these phages are highly specific to *Salmonella* Typhi.

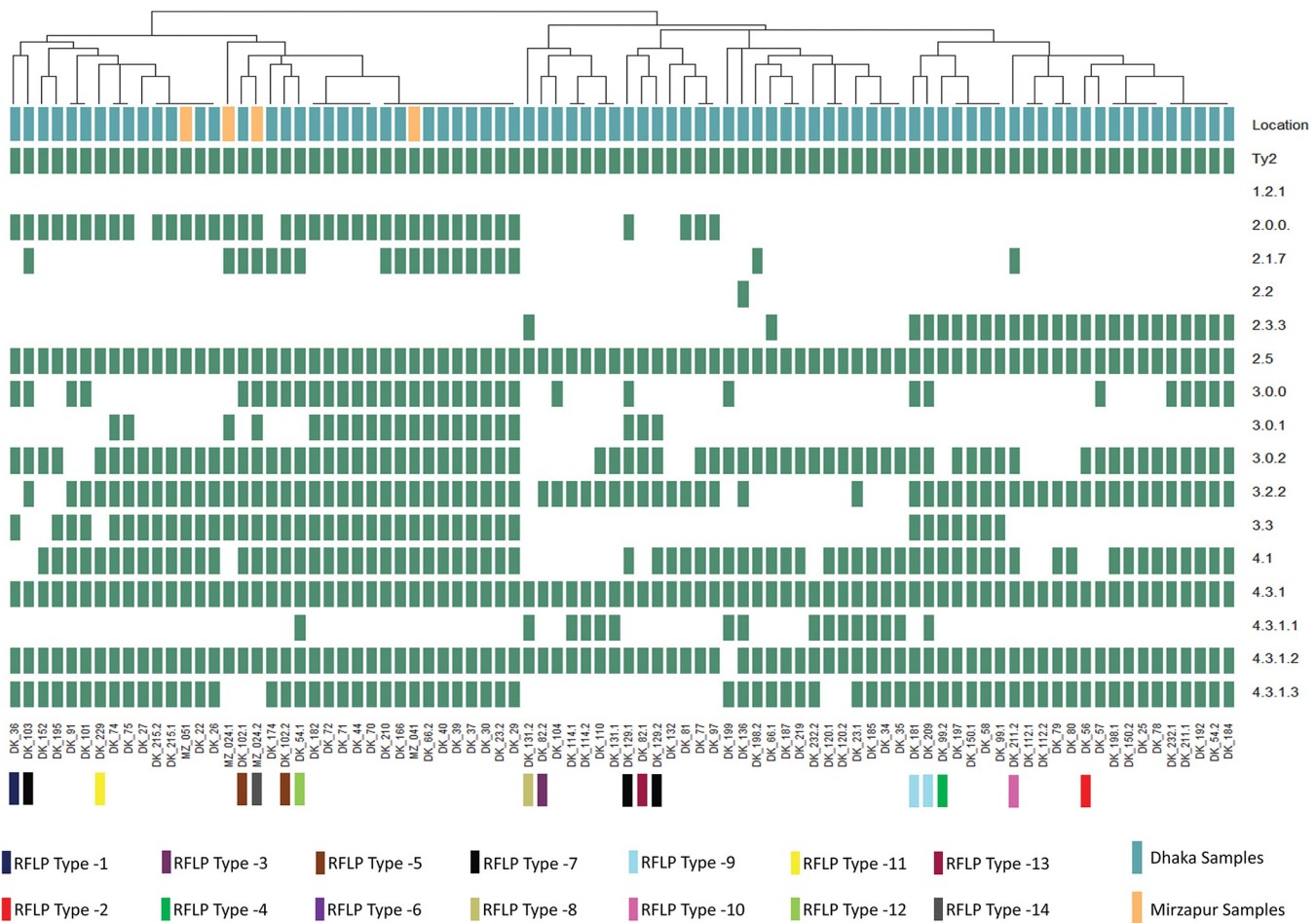

**Fig 2. Hierarchical clustering of the activity spectra of the 86 bacteriophages isolated in the study.** All isolated phages were spotted on *Salmonella* Typhi isolates belonging to different genotypes (Y-axis) circulating in Bangladesh. Green represents activity/plaque formation, white represents no activity/no plaque formation. The phages on the X-axis are labelled based on the site of isolation. Phages from Mirzapur are labelled as MZ, and From Dhaka are labelled as DK. Multiple phages isolated from the sample are indicated with .1 and .2 at the end of the label. The heatmap was made using the R package gplots. The RFLP types determined by digestion with XbaI/NheI are also shown. Attenuated *Salmonella* Typhi strain Ty2, BRD948, is denoted as Ty2.

Next, to understand the diversity of the phages collected, we tested the killing spectrum of the 86 phages against a panel of 16 *Salmonella* Typhi strains each representing a different genotype circulating in Bangladesh [20] (Fig 2). The phages showed a diverse killing spectrum and could be grouped into 48 clusters based on their killing activity. This demonstrated that the circulating *Salmonella* Typhi strains in Bangladesh are not equally susceptible to all environmental phages; some are more susceptible than others. In addition, phages with different plaque morphologies from the same sample often displayed (for example DK_23.1 and DK_23.2) different spectra, indicating that multiple phages can circulate in the same water body at the same time.

To investigate the genetic diversity of the phages, we optimized a restriction fragment length polymorphism (or RFLP) assay using NheI and XbaI on 20 different phages with different killing activity [24,25]. In total, we observed 14 RFLP types based on fragment size from 17 phages that gave a clear banding pattern (Figs 2 and S1). The fragment sizes ranged from 1,000 bp to 15,000 bp. Two phages that had similar killing spectra (DK_181 and DK_209) showed similar banding patterns in the RFLP assay. Interestingly, two pairs of phages (DK_102.1/

DK_102.2; RFLP type-5 and DK_129.1/DK_129.2; RFLP type-7) showed different morphology and activity spectra but had the same RFLP type. Taken together, the distinct RFLP types observed confirmed the genetic diversity of the isolated phages.

## Discussion

Our pilot study shows that detection of bacteriophages specific to *Salmonella* Typhi may be a rapid environmental surveillance method to understand the presence of typhoid fever in the community. 33% of environmental samples collected in Dhaka contained phages, where blood culture positivity was 5%; by comparison, 3% of environmental samples collected in Mirzapur contained phages, where blood culture positivity was 0.05%. Of all the sources of water collected, 75% (52/69) of samples were collected from sewage indicating that wastewater surveillance is well-suited for monitoring typhoid fever. In Dhaka, a city with high burden of typhoid fever, high phage positivity was also seen in water samples collected from lakes (56%) and rivers (40%).

Given the minimal resources required for undertaking environmental phage surveillance, it can be readily rolled out in resource-constrained settings and can complement existing surveillance strategies. Sample processing primarily includes collection bottles and/or tubes, syringes with 0.22 μm syringe filters, petri dishes, media for bacterial growth, an incubator, a centrifuge, pipettes, and the laboratory strain BRD948 of *Salmonella* Typhi (See Methods). The cost for consumables for each sample is less than USD 10 in Bangladesh (this might vary slightly based on location). Such low costs of sample collection and easy processing means that phage surveillance is more scalable than PCR-based molecular methods, which are both resource and expertise intensive. Furthermore, the stability of phages [26] in water means that samples can be collected from across the country and tested at any location without loss of quality of the results obtained. In contrast, environmental DNA degrades fast, and thus PCR based surveillance typically requires transport of refrigerated or frozen samples.

Limited work has been done in investigating the role that phages play in the ecology of *Salmonella* Typhi. The abundance and diversity of Typhi phages in terms of the killing spectrum and the genetic sequences highlight that Typhi bacteriophages are likely to play an important role in determining the spread and seasonality of *Salmonella* Typhi. Additionally, certain genotypes of *Salmonella* Typhi (such as 1.2.1, 2.2 and 4.3.1.1) are more resistant to phages vs others (such as 4.3.1.2, 4.3.1 and 2.5). The molecular basis of the observed differences in phage resistance amongst different genotypes may be due to differences in receptor sequences and/or modifications, or presence of phage defense systems (such as CRISPR-Cas). Future studies addressing these questions may be helpful in determining the impact phages have on the circulating *Salmonella* Typhi population [27,28]. Additionally, with rising antimicrobial drug resistance, renewed research on Typhi phages might provide alternate therapeutics for clinical development.

The results in this study should be interpreted within the context of the following limitations. First, the number of samples obtained does not fully represent the number of water bodies present in Dhaka or Mirzapur. Second, no water sampling could be conducted from July-August during the study period due to monsoon-associated floods. Third, all phage amplification steps were done using *Salmonella* Typhi strain BRD948, and hence phages that do not infect this strain were missed. In future studies, a derivative of this strain could be developed with no restriction modification or phage defense systems. This would increase the number of phages identified. Fourth, sewage samples were underrepresented in Mirzapur due to lack of a sewage system in many rural areas. Finally, comparison with the PCR-based assays will be required to identify the specificity and sensitivity of phage detection. Expansion of this study

to other regions of Bangladesh where clinical data is available will help in resolving some of these limitations. It would also be helpful to examine temporal trends in phage positivity and how these trends correlate with the seasonality of typhoid fever.

In summary, in this study, we propose a simple, cost-effective, and scalable method for conducting environmental surveillance for typhoid fever. The method uses standard microbiological laboratory infrastructure and techniques to detect Typhi-specific bacteriophages. Environmental phage surveillance can be used to estimate typhoid in other countries, including in Sub-Saharan Africa where limited epidemiological data on typhoid fever is available [29,30]. Environmental phage surveillance may be also applied to map routes of disease transmission by focusing on wastewater/sewage facilities in the region. This tool, combined with traditional blood culture surveillance [31], can generate community-level data to evaluate the impact of interventions including the introduction of TCV, water improvement projects, and sanitation and hygiene systems.

## Supporting information

**S1 Fig. Restriction fragment length polymorphism (RFLP) assay on 20 phages.** DNA was isolated from phages and was digested with NheI/XbaI. The images before (top left) and after (top right) lane and band identification. The table (bottom) shows the predicted band sizes. Based on the band patterns, different phages were assigned a RFLP-Type.
(TIFF)

## Author Contributions

**Conceptualization:** Jason R. Andrews, Senjuti Saha.

**Formal analysis:** Yogesh Hooda.

**Funding acquisition:** Stephen P. Luby, Jason R. Andrews, Samir K. Saha, Senjuti Saha.

**Investigation:** Yogesh Hooda, Shuborno Islam, Rathin Kabiraj, Himadree Sarkar, Senjuti Saha.

**Methodology:** Yogesh Hooda, Shuborno Islam, Rathin Kabiraj, Hafizur Rahman, Himadree Sarkar, Kesia E. da Silva, Senjuti Saha.

**Project administration:** Rathin Kabiraj, Hafizur Rahman, Senjuti Saha.

**Resources:** Samir K. Saha.

**Supervision:** Yogesh Hooda, Samir K. Saha, Senjuti Saha.

**Validation:** Yogesh Hooda, Shuborno Islam, Kesia E. da Silva, Jason R. Andrews.

**Visualization:** Yogesh Hooda, Shuborno Islam, Himadree Sarkar, Rajan Saha Raju, Senjuti Saha.

**Writing – original draft:** Yogesh Hooda, Shuborno Islam, Senjuti Saha.

**Writing – review & editing:** Yogesh Hooda, Shuborno Islam, Rathin Kabiraj, Kesia E. da Silva, Stephen P. Luby, Jason R. Andrews, Samir K. Saha, Senjuti Saha.

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
