## [Decision Letter · Decision Letter 0]

15 Aug 2023

Dear Dr. Saha,

Thank you very much for submitting your manuscript "Old tools, new applications: use of environmental bacteriophages for typhoid surveillance and evaluating vaccine impact" for consideration at PLOS Neglected Tropical Diseases. As with all papers reviewed by the journal, your manuscript was reviewed by members of the editorial board and by several independent reviewers. The reviewers appreciated the attention to an important topic. Based on the reviews, we are likely to accept this manuscript for publication, providing that you modify the manuscript according to the review recommendations. Take special consideration on comments from Reviewer #2 (Comment 3) about the use of different molecular techniques. 

Sincerely,

Elsio A Wunder Jr, DVM, Ph.D.

Section Editor

Reviewer's Responses to Questions

**Key Review Criteria Required for Acceptance?**

**Methods**

-Are the objectives of the study clearly articulated with a clear testable hypothesis stated?

-Is the study design appropriate to address the stated objectives?

-Is the population clearly described and appropriate for the hypothesis being tested?

-Is the sample size sufficient to ensure adequate power to address the hypothesis being tested?

-Were correct statistical analysis used to support conclusions?

-Are there concerns about ethical or regulatory requirements being met?

Reviewer #1: This study is well designed and its simplicity enviable. All questions and objections are clearly stated. Please see the comments to the author's for a more detailed analysis and specific recommendations and questions.

Reviewer #2: (No Response)

Reviewer #3: Methodology is appropriate, but need citations.

**Results**

-Does the analysis presented match the analysis plan?

-Are the results clearly and completely presented?

-Are the figures (Tables, Images) of sufficient quality for clarity?

Reviewer #1: The analysis conducted is appropriate and the results are well stated. Figure one could be more clear, see below for details.

Reviewer #2: (No Response)

Reviewer #3: The results brings, in fact, surveillance strategy for typhoid surveillance, but the evaluation of vaccine impact is poorly assessed.

**Conclusions**

-Are the conclusions supported by the data presented?

-Are the limitations of analysis clearly described?

-Do the authors discuss how these data can be helpful to advance our understanding of the topic under study?

-Is public health relevance addressed?

Reviewer #1: The conclusions are supported by the data, the authors consider the limitations of their proposed surveillance technique, and implications and utility are thoroughly discussed.

Reviewer #2: (No Response)

Reviewer #3: The conclusions are in accordance with study data.

**Editorial and Data Presentation Modifications?**

Reviewer #1: Only minor changes are needed in terms of editing and data presentation. Based on my questions below, the authors may elect to provide changes not explicitly requested here.

Reviewer #2: (No Response)

Reviewer #3: I recommend to check image quality/resolution.

**Summary and General Comments**

Reviewer #1: Review of Hooda et al. Old tools, new applications: use of environmental bacteriophages for typhoid surveillance and evaluating vaccine impact

Critical to eradicating vaccine-preventable diseases is knowing where to deploy vaccines. This report provides a cheap, reproducible, and tractable method for monitoring the infection burden of Salmonella enterica Typhi – the causative agent of typhoid. While infectious of typhoid are globally distributed, there is a particular prevalence in lower resourced countries, furthering the need for cheap methods for monitoring regional infection burden. The techniques used in this manuscript are well explained and easily reproducible and the experiments used to validate the methods employed herein are robust and convincing. 

Based on the above criteria, I support publication of this report in PLOS Neglected Tropical Diseases with only minor changes. Below I have provided comments and suggestions on how to improve the manuscript as well as a suggested control that would substantially increase the impact of the work.

Experimental Suggestion:

1. A negative control looking at the presence (or hopefully the absence) of S. Typhi phages from water samples in a region with no confirmed typhoid infections would substantially increase the impact of this work and make the method extremely convincing.

Major Comments and Suggestions

1. Figure one is difficult to interpret, in particular the fact that the inset maps have the same background is confusing. Does the Dhaka area include Mirpur and the other named location? Are all sampling locations found off of the N4 (assuming that’s what the pink line is)? Do the inset boxes have the same distance scaling?

2. It is the author’s intent for this method to be generalizable and used for surveillance, but they mention as a limitation that only phages that infect their isolation host (Ty2) will be isolated with this method. Therefore, it is necessary for the authors to justify using Ty2 as the sole isolation host. Is it known that a large variety of phages can infect this strain? Does this strain lack phage defenses as many other common phage isolation hosts due (e.g. E. coli C or S. aureus RN4220)? Do all the clinically relevant strains infect this strain?

Minor Comments and Suggestions

1. The introduction gives focus to wastewater surveillance. Why were only three sewage samples taken from Mirzapur? Would the same high degree of positive samples be expected should the number of samples be increased?

2. There are a few small typographical errors that would be caught by a careful re-reading such as “The burden of culture-confirmed cases of typhoid fever in Mirzapur in 100-fold lower,” on line 173.

Reviewer #2: The manuscript titled “Old tools, new applications: use of environmental bacteriophages for typhoid surveillance and evaluating vaccine impact” suggests that it is possible to detect the presence of Salmonella Typhi bacteriophages in environmental water samples as an alternative to the traditional blood culture monitoring method to determine the typhoid burden. The main aim of the study was to determine the relationship of bacteriophages to the typhoid burden in the environment and to examine the usability of this information to aid typhoid control. The authors state that the environmental bacteriophage surveillance method can also be used in other epidemiological data-deficient regions due to its easy applicability, low cost, and speed of data generation. The authors also discussed that isolated bacteriophages have potential to be used not only as a monitoring tool but also for phage therapy in the future. Overall, the article is well written and the methodology is solid. The authors detailed the methods they used to collect and analyze the water samples. The authors of the study summarized the research findings in a clear and understandable way and emphasized the practical importance of their findings. They also discuss the limitations of the study and suggest directions for future research.

However, the article could be improved in several ways. Please find my comments below:

1- Firstly, the authors can provide more information about the Salmonella Typhi strains used in the study. In particular, it should be explained in a few sentences why Salmonella Typhi strain Ty2 was selected for resource enrichment and all phage amplification steps. This information is important for the interpretation of results, as different strains of Salmonella Typhi may be more or less susceptible to bacteriophages.

a) Example sentence 1; Salmonella Typhi strain Ty2 was selected as indicator bacteria in all phage amplification steps. Because the Ty2 strain is a strain that is weak or lacking in CRISPR-Cas and/or restriction endonucleases that are associated with phage defense systems.

b) Example sentence 2; Salmonella Typhi strain Ty2 was selected as indicator bacteria in all phage amplification steps. Because Ty2 is both a laboratory strain and a strain with low pathogenicity, it was preferred in terms of work safety.

Please add a sentence or two to the methodology section or discussion section of the manuscript to explain why you chose the Salmonella Typhi strain Ty2 (as in the two examples above).

2- The Methods section describes the methods used. However, I think some steps should be explained in more detail and clearly. This may help other researchers repeat similar studies.

a) Please include the centrifuge pattern throughout the text or calculate centrifuge speed in "g" instead of "rpm".

b) What is the titer of the phages applied on the bacteria in the spot-test experiment conducted under the title of "Activity spectra of Salmonella Typhi bacteriophages"? If known, please indicate the applied phage titers in one sentence (For example, 2 µl of phage stocks with titers ranging from 108 Pfu/ml or 106-108 Pfu/ml were dropped on bacterial grass).

c) Page 8, lines 151-155: 2 µl dilutions of isolated bacteriophages were dropped onto bacteria using double layer methodology. If 2 µl dilutions in the sentence mean the direct filtrate of each phage clone, please correct the dilution expression as filtrate. If it is meant to be dilutions prepared from phage filtrate, indicate how many fold it was diluted and used.

d) Page 8, line 151 and Page 11, line 205: It is stated that 14 Salmonella Typhi strains representing different genotypes circulating in Bangladesh were used to understand the bacteriophage diversity. However, the number of bacteria tested on the X-axis of picture 2 shown in the results is more than 14. There is an inconsistency between picture 2 and the sentences. Please check and fix the incompatibility.

3- To determine the diversity of 86 bacteriophages isolated in the study, the authors tested the bacteriophages against a panel of 14 Salmonella Typhi strains, each representing a different genotype, and grouped the phages into 48 clusters according to their killing activity..

a) Adjusting different phages to the same titer and testing under a single experimental condition makes the results more reliable and comparable. This helps to more accurately assess the true killing capacity of different phages. The titer of the 86 phages tested in the study was not predetermined (or not specified in the article). For this reason, the grouping approach based on the killing spectra specified in the manuscript may offer limited perspective.

b) The lytic spectrum data indicated by the article is important in grouping isolated bacteriophages, and grouping phages based on a particular killing activity may offer a way to initially understand different activity levels. However, if the authors have the means and technical equipment, my recommendation is to use more detailed molecular techniques, such as restriction fragment length polymorphism (RFLP) analysis or randomly amplified polymorphic DNA (RAPD)-PCR techniques, which may be helpful in grouping phages and assessing their genetic variation more precisely. .

c) Also, grouping phages with RFLP or RAPD-PCR techniques has other advantages. If the authors decide to sequence these phages for further molecular characterization in the future, phages that are identical at the genome level will be less likely to be sequenced multiple times. This will reduce the sequencing cost of the authors.

d) For the benefit of the authors, I include below the DOI numbers of two studies that performed phage typing using RFLP analysis and RAPD-PCR techniques.

https://doi.org/10.1111/j.1574-6968.2011.02342.x

https://doi.org/10.1016/j.virusres.2023.199049

Conclusion:

Overall, the article is well written, organized and provides solid evidence to support the research findings. The results of the article are interesting and important, showing that Salmonella Typhi bacteriophages can play an important role in determining the spread and seasonality of bacteria. These findings will be valuable for future research to develop new methods to monitor and control Salmonella Typhi. I believe that with the corrections to be made by paying attention to the points I mentioned above, the article will be accepted as a better understandable and valuable contribution.

Reviewer #3: The paper brings a relevant strategy for monitoring an emergent disease and could be very usefull to avoid burdens of typhoid fever burdens, specially where material and logistical resources are scarce.

PLOS authors have the option to publish the peer review history of their article (what does this mean?). If published, this will include your full peer review and any attached files.

Reviewer #1: No

Reviewer #2: Yes: Abdulkerim Karaynir

Reviewer #3: Yes: Thiago Accioly

Figure Files:

Data Requirements:

Reproducibility:

References

---

## [Editor Report · Decision Letter 1]

27 Nov 2023

Dear Dr. Saha,

We are pleased to inform you that your manuscript 'Old tools, new applications: use of environmental bacteriophages for typhoid surveillance and evaluating vaccine impact' has been provisionally accepted for publication in PLOS Neglected Tropical Diseases.

Best regards,

Elsio A Wunder Jr, DVM, Ph.D.

Section Editor

Elsio Wunder Jr

Section Editor

---

## [Editor Report · Acceptance letter]

22 Jan 2024

Dear Dr. Saha,

We are delighted to inform you that your manuscript, "Old tools, new applications: use of environmental bacteriophages for typhoid surveillance and evaluating vaccine impact," has been formally accepted for publication in PLOS Neglected Tropical Diseases.

Best regards,

Shaden Kamhawi

co-Editor-in-Chief

Paul Brindley

co-Editor-in-Chief
